# Nerve Growth Factor: A Potential Therapeutic Target for Lung Diseases

**DOI:** 10.3390/ijms22179112

**Published:** 2021-08-24

**Authors:** Piaoyang Liu, Shun Li, Liling Tang

**Affiliations:** 1Key Laboratory of Biorheological Science and Technology, Ministry of Education, College of Bioengineering, Chongqing University, Chongqing 400044, China; 201919021042@cqu.edu.cn; 2Department of Immunology, School of Basic Medical Sciences, Chengdu Medical College, Chengdu 610500, China; 3Non-Coding RNA and Drug Discovery Key Laboratory of Sichuan Province, Chengdu Medical College, Chengdu 610500, China

**Keywords:** nerve growth factor, lung, coronavirus disease 2019, pathway

## Abstract

The lungs play a very important role in the human respiratory system. However, many factors can destroy the structure of the lung, causing several lung diseases and, often, serious damage to people’s health. Nerve growth factor (NGF) is a polypeptide which is widely expressed in lung tissues. Under different microenvironments, NGF participates in the occurrence and development of lung diseases by changing protein expression levels and mediating cell function. In this review, we summarize the functions of NGF as well as some potential underlying mechanisms in pulmonary fibrosis (PF), coronavirus disease 2019 (COVID-19), pulmonary hypertension (PH), asthma, chronic obstructive pulmonary disease (COPD), and lung cancer. Furthermore, we highlight that anti-NGF may be used in future therapeutic strategies.

## 1. Introduction

The lung is one of the most important respiratory organs. When people begin to breathe, the lung realizes the gas exchange between the body and the external environment to maintain human life activities [1]. It is also an extremely complex organ in which the coordinated work of dozens of different cells ensures normal functioning. Because of its complex structure, it is difficult to diagnose many lung diseases. There are many lung-related diseases, such as pulmonary fibrosis (PF) [2], pulmonary hypertension (PH) [3,4], asthma [5], chronic obstructive pulmonary disease (COPD) [6], and lung cancer [7,8].

Nerve growth factor (NGF), a member of the neurotrophin family, can increase the number of sympathetic and sensory neurons associated with the development of the nervous system [9]. NGF is also extensively expressed in lung tissues. In recent decades, interest in the role of NGF in lung diseases has increased. Although the pathology of each lung disease may be different, in diverse microenvironments, NGF participates in the occurrence and development of lung diseases, like PF, bronchial asthma, and lung cancer, through changing protein expression levels and mediating cell function [10].

In this review, we summarize the functions of NGF and highlight its effects on lung diseases. NGF is a latent therapeutic target and anti-NGF drugs may improve lung damage.

## 2. NGF

### 2.1. NGF and Its Receptors

NGF consists of 118 amino acids, and its molecular weight is 130 kDa [11]. NGF is a member of the neurotrophin family [2,12] and is related to neurotrophins in the peripheral nervous system [13] and central neural system. NGF contains three subunits: α, β, and γ [13].

NGF activates two sorts of receptors: the high-affinity tropomyosin-related kinase A receptor (TrkA) and the low-affinity p75 neurotrophin receptor (p75NTR) [14]. The TrkA receptor, a 140-kDa transmembrane protein, is encoded by a proto-oncogene on chromosome 1. Due to its intrinsic tyrosine kinase activity, TrkA selectively binds to NGF. When TrkA is activated, NGF induces cell proliferation, cell differentiation, and cell survival. In addition, TrkA inhibits apoptosis, increases the excitability of neurons, and induces mediator release. The p75NTR receptor, encoded by a gene situated on chromosome 17, is a 75-kDa glycoprotein. It combines all neurotrophins and the prosoma (pro) of neurotrophins, most notably pro-NGF. After pro-NGF binds to p75NTR, apoptosis is activated (Figure 1).

### 2.2. NGF and the Lung

#### 2.2.1. NGF in the Lung

NGF and its receptors are expressed by many lung cells, including epithelium, smooth muscle, fibroblasts, and vascular endothelium. Additionally, B and T lymphocytes, as well as dendritic cells, monocytes, mast cells, and macrophages, release high levels of NGF [15]. Through immunolocalization, NGF is also found in the bronchial epithelium, smooth muscle cells, and fibroblasts in the bronchi of healthy subjects. Furthermore, basal NGF levels could be detected in bronchoalveolar lavage fluid from healthy subjects [16]. This suggests that NGF is also expressed in human lungs under healthy conditions. Accordingly, NGF has the potential to affect lung structure and function.

The expression levels of NGF and its receptors TrkA and p75NTR are increased in fibrotic lung tissue compared to normal tissue [17]. Monocrotaline-induced PH is related to increased NGF levels in lung tissue [18]. NGF, a critical mediator in neuro-immune mechanisms of asthma, aggravates inflammation in asthma and airway remodeling [19]. NGF boosts angiogenesis in tumor tissue, including non-small cell lung cancers (NSCLCs) [20]. Serum levels of NGF are also extremely high in COPD subjects [14]. This suggests that PF, PH, asthma, NSCLC, and COPD are associated with NGF (Figure 2). In the following sections, the relationship between NGF and these lung diseases is described in detail.

#### 2.2.2. NGF and Affected Pathways in Lung Diseases

A recent study provides evidence that many factors, but not age or sex, are related to NGF levels [21]. MicroRNAs (miRNAs, miRs) are small non-coding RNAs with a length of about 22 nucleotides. MiRNAs control protein expression via inducing mRNA degradation or inhibiting mRNA translation. MiRNAs are involved in various biological processes, like proliferation, development, differentiation, and apoptosis [22]. NGF may regulate the expression of diverse genes by controlling miRNA expression, including those whose functions and processes are known to be related to NGF [23]. In the lung, NGF may have pro-fibrotic effects. In lung fibroblasts, the pro-fibrotic effects of NGF are inhibited because transforming growth factor-β1 (TGF-β1)-induced *miR-455-3p* limits the production of NGF [24]. Moreover, the increase in the levels of NGF and other markers of nicotine-induced airway remodeling is negatively regulated by *miRNA-98* [25]. Overexpression of *miR-221* is related to maximal downregulation of NGF and its TrkA receptor at both the mRNA and protein levels [26]. To explore the molecular mechanisms underlying the effects of NGF in the lung, in this review we provide a detailed account of recent advances in research on NGF signaling.

## 3. NGF and Lung Diseases

### 3.1. NGF and PF

PF, which can be idiopathic or secondary to various medical conditions, is characterized by excessive scarring of the lungs. Alveolar epithelial cell injury, destruction of the alveolar–capillary membrane, and abnormal vascular repair, alongside tissue remodeling and extracellular matrix (ECM) deposition, have been proposed as probable pathogenic mechanisms [27]. The lung architecture and function are destroyed and alveolar gas exchange is disrupted when PF occurs. PF leads to hypoxemia, dyspnea, exercise intolerance, and finally death. During the past few decades, the care of lung disease patients has been improved by medical advances, but the incidence and mortality of PF have scarcely improved. Novel therapies to ameliorate the outcomes of PF are urgently required [28].

NGF boosts the growth and ramification of peripheral nerve fibers. The axons of peripheral nerve fibers are wrapped in non-myelinating glial cells named satellite cells. These fibroblast-associated cells physically sustain the axon by synthesizing collagen fibers parallel to the length of the nerve fibers. Therefore, one of the results of an increase in NGF levels is an increase in mesenchymal cells and excessive deposition of ECM. These are signs of fibrosis [2]. In excessive innervation tissues in NGF-injected mice or NGF-overexpressing transgenic mice, the nerve bundles are full of collagen fibers and satellite cells. In the kidney, high NGF serum levels could worsen renal fibrosis [29]. In thioacetamide-induced liver fibrosis, NGF is a critical moderator of the stress-induced fibrogenesis signaling pathway by activating p75NTR, increasing liver damage [30]. In fibrotic lung tissue, NGF expression was increased on immunohistochemical detection compared to normal lung tissue [17]. Furthermore, under basal conditions, TrkA is expressed in human lung fibroblasts (Figure 3) [17]. After NGF treatment, these cells also increase NGF production. NGF promotes the migration of fibroblasts. It also enhances the migration of human fetal lung fibroblasts towards fibronectin and platelet-derived growth factor (PDGF) [31]. The chemotaxis and chemokinesis of these cells seem to be affected by NGF. Thus, this shows that NGF participates in the process of fibrosis. NGF may be a potential treatment target for fibrosis, including PF. Anti-NGF may improve the symptoms of PF.

Coronavirus disease 2019 (COVID-19) is an acute respiratory contracted disease. It is caused by infection with a novel coronavirus, named severe acute respiratory syndrome coronavirus 2 [32], which has rapidly spread in the world. As of June 6, 2021, there were more than 172 million confirmed cases and nearly 4 million deaths around the world [33]. Many COVID-19 imageology and pathology studies have been published. Through chest computed tomography (CT) scans, fibrotic changes have been found in COVID-19 patients. For example, Zhou et al. reported fibrotic changes on the chest CT scan of 21 out of 62 patients [34]. Similarly, in a study of 63 patients by Pan et al., fibrotic changes were seen in 11 patients during acute illness [35]. These imaging findings are supported by autopsy and puncture reports. Liu et al. reported gross anatomy results of the world’s first case of COVID-19 [36]. Subsequently, Yao et al. reported three cases of puncture pathological results [37]. They all found different degrees of PF in the pathological examinations. Similarly, fibroblastic proliferation and deposition of ECM and fibrin in the alveolar space were also observed in four patients who died of COVID-19 pneumonia [38]. Meanwhile, several studies [39,40,41] have reported that the levels of cytokines involved in PF are increased during COVID-19 development. For instance, interleukin 1 (IL-1) is a regulatory molecule of the fibrotic response in PF. Increased secretion of IL-1 has also been observed in COVID-19 patients [40]. Therefore, we speculate that the treatment of PF could be beneficial for COVID-19 patients.

Though the relationships between NGF and COVID-19 have been not described thus far, NGF may also be a potential therapeutic target for COVID-19 because NGF is an emerging target in PF.

### 3.2. NGF and PH

PH is a progressive and chronic disease. It causes right heart failure and if untreated it leads to death [42]. The pathobiology of PH is extremely complicated and involves many factors, such as an imbalance between vasoconstriction and vasodilatation, smooth muscle cell proliferation, and vascular inflammation [43]. Current treatment strategies can improve some symptoms but cannot cure them.

NGF is a growth factor that plays a crucial role in the pathophysiology of PH [18], especially in overreaction, remodeling, and pulmonary vascular inflammation [44]. Studies in animals and humans have shown that NGF promotes (i) the proliferation and migration of vascular cells and (ii) high reactivity or secretion of pro-inflammatory cytokines in pulmonary arteries [45,46,47]. In addition, a previous study demonstrated that the proliferation and migration of airway smooth muscle cells are stimulated through NGF and its receptors [45]. The receptors and signaling pathways involved in these phenomena differ per cell type. These processes depend only on the TrkA receptor in smooth muscle cells of human pulmonary arteries (Figure 3) [44]. At the same time, it has been found that the beneficial effects of PDGF or Rho-associated protein kinase (ROCK) inhibition on monocrotaline-induced PH could partially be mediated by a reduction in NGF signal transduction [48]. NGF probably promotes activation of one of the receptor tyrosine kinase or Ras homolog family member A (RhoA)–ROCK pathways (Figure 3). However, few data on the possible links between NGF and other molecular pathways that have an immediate or latent effect on the development of PH are available.

In short, these findings indicate that NGF may serve as a target of special interest in therapeutic strategies to treat PH in the future. The use of NGF-blocking agents is regarded as a future treatment of choice for PH [45].

### 3.3. NGF and Asthma

Asthma is a chronic, hereditary, and heterogeneous respiratory inflammatory disease characterized by reversible obstruction and bronchial hyperresponsiveness. Asthma results from an immune response, mediated by T-helper cells type 2 (Th2), to immunological, neurogenic, chemical, physical, and environmental stimuli [49,50,51]. The disease symptoms of most patients can be controlled by simultaneously inhaled corticosteroids and bronchodilators [50,52]. Although the symptoms of 90–95% of patients can be improved, the rest suffer from refractory asthma [53]. Therefore, it is necessary to further research the mechanisms causing the development of chronic inflammation and bronchospasm to develop novel and effective treatments to treat this disease [54].

Many mediators produced by mast cells and eosinophils play a significant role in the pathophysiology of asthma. NGF is generated by eosinophils [55,56] and mast cells [55,57]. At the same time, a study has shown that the level of NGF is related to the number of eosinophils, which are the main effector cells of asthma [58]. Therefore, NGF has also been deemed a significant factor in the pathogenesis of asthma [59]. Hahn et al. have suggested that the high expression level of NGF enhances eosinophilic vitality, activating the eosinophils in the airways resulting from allergens and partaking in the evolvement of persistent airway hyperresponsiveness (AHR) in mice [60]. Other studies have also reported that in asthmatic rat cells, the increase of NGF mRNA expression is always followed by the appearance of high levels of alpha smooth muscle actin (α-SMA) mRNA. It also accompanies the continuous immune response of fibroids and strengthens smooth muscle cells [61]. Using an ovalbumin (OVA)-induced asthma rat model, it has been found that NGF aggravates inflammation, AHR, and airway remodeling via the Th2 immune response and by increasing matrix metalloproteinase-9 (MMP-9) expression (Figure 3) [62]. The effect of NGF in asthma by the Th2-mediated immune response has been confirmed by Qin et al. [63]. In another study, NGF was shown to be a key factor in lung inflammation and Th1/Th2 balance in asthma after respiratory syncytial virus infections [64]. Asthma patients demonstrate increased levels of NGF after allergenic bronchial provocation, and NGF levels are also closely related to the severity of the disease [58,65].

Ogawa et al. found that NGF-targeted small interfering RNA reduces the expression of NGF in bronchial and alveolar epithelial cells in a dust mite-induced chronic asthma mouse model [66]. Later, other researchers investigated the impact of NGF inhibition on AHR and other asthma phenotypes in an asthma mouse model. Inhibition of NGF through NGF-targeting short hairpin RNAs also appeared to reduce the severity of asthma phenotypes [67]. Research on the pathogenesis of asthma in OVA-sensitized mice furnishes proof that NGF blockade reduces airway inflammation [68]. It has been verified that NGF blockade inhibits airway allergic inflammation via regulating the balance of Th1 and Th2 responses of T cells in experimental asthma models [69]. The high expression of tissue inhibitors of NGF and metalloproteinase-1 (TIMP-1) and the correlation between parameters of asthma patients show that there may be a relationship between NGF and TIMP-1, which may play a significant role in the pathogenesis of asthma (Figure 3). Renz et al. have shown that in asthma, NGF also promotes airway remodeling [65].

All of these studies show that NGF is a mediator of asthma, and NGF blockade improves some symptoms. Several NGF-activated pathways in asthma have been identified.

Rho family proteins were the first proteins to be cloned in the Ras superfamily. RhoA belongs to the Rho GTP enzyme family, and it is a critical factor in cell signal transduction pathways. Additionally, in intracellular signal transduction, it acts as a bridge [70,71,72]. In OVA-sensitized asthmatic guinea pigs, the expression of RhoA is increased. Additionally, its expression is inhibited by Rho kinase inhibitors; in the case of inhibition, airway inflammation and systole of smooth muscles are improved [73]. It was also found that OVA resulted in an imbalance in the ratio of Th1 and Th2 cells in mice. This leads to airway inflammation and increases the expression of NGF and RhoA, thereby increasing the activity of its downstream effector ROCK. ROCK phosphorylates myosin light-chain phosphatase and induces airway smooth muscle contraction. OVA potentially causes asthma via the Rho–ROCK pathway (Figure 3). NGF partakes in the pathogenesis of an OVA-sensitized asthma mouse model. Therefore, anti-NGF possibly improves AHR and relieves asthma attacks in mice via downregulating the RhoA pathway (Figure 3) [74].

TGF-β1 is the most powerful fibrotic factor, and its main function in the airway is to promote the synthesis of ECM. Recent studies have confirmed that TGF-β1 is a critical cytokine in the process of airway remodeling [75]. SMAD3 is an intracellular protein. SMAD3 is essential for transmitting TGF-β1 signals from the cell surface to the nucleus, where it promotes target gene transcription [76]. In the kidney, it was found that this signaling pathway also regulates NGF. NGF activates epithelial–mesenchymal transition (EMT) markers through its receptors TrkA and p75NTR, thereby transcriptionally upregulating the expression and secretion of TGF-β1 and enhancing cell motility. Interestingly, TGF-β1 knockdown prevents TGF-β1–SMAD pathway activation and NGF-induced upregulation of EMT markers, and pretreatment with anti-NGF reverses the nuclear translocation of the p-SMAD3–SMAD4 complex [29]. In OVA-induced animal models, treatment with anti-NGF microspheres (NANM) significantly reduces airway remodeling. It also decreases the mRNA expression of TGF-β1 and regulates the expression of the downstream marker p-SMAD3. This shows that NANM may inhibit airway remodeling in asthma animal models by regulating the signaling pathway of TGF-β1–SMAD3 (Figure 3) [77].

These data have indicated that NGF may worsen airway inflammation, hyperresponsiveness, and remodeling. Therefore, NGF could be a therapeutic target in asthma.

### 3.4. NGF and COPD

COPD is a disease of irreversible and progressive airway inflammation, bronchial blockage, and lung parenchymal sabotage [78,79]. It is one of the main causes of morbidity and mortality around the world. COPD is characterized by a very complex pathogenesis, involving the release of multiple cytokines [80,81] and NGF [14,82] and its receptor regulation. Different inflammatory mediators, such as cytokines and chemokines, participate in the treatment of COPD.

During COPD, the levels of monocyte chemoattractant protein 1 and pro-inflammatory cytokines such as tumor necrosis factor-α gradually increase [80,81,83,84,85], but few data about the participation of NGF are available [82,86,87,88]. Some other studies have shown that NGF and its receptors are involved in high bronchial responsiveness, like asthma [15,89,90,91] and allergic rhinitis [56]. But the effect and importance of the NGF–NGF receptor system in COPD are still unclear.

Stabile et al. found that the severity of COPD was correlated with increased NGF serum levels. In COPD patients, the serum levels of NGF are increased threefold (mild-to-moderate) to sixfold (severe-to-very severe) [14]. Previous in vitro and in vivo studies have shown that the activation of TrkA promotes inflammation and tissue remodeling. Particularly, in rodent models of asthma, pretreatment with TrkA blocking agents reduced the reaction to allergen sensitization [19]. Moreover, increased expression of TrkA in alveolar macrophages [14,92,93] and bronchial epithelial cells [94,95,96,97] has been found in rodent models or patients of COPD. Though p75NTR is expressed in smooth muscle cells, epithelial cells, and fibroblasts in the respiratory system and it participates in asthma and inflammatory responses [12], in the different stages of COPD, TrkA and p75NTR exhibit different expression levels.

Because the bioactivity of NGF is regulated by its receptors TrkA and p75NTR (Figure 3), NGF is continuously involved in the progressive inflammation process of COPD. It may be a therapeutic target for COPD.

### 3.5. NGF in Lung Cancer

Lung cancer is the main cause of cancer-related death around the world, and its incidence is still increasing [98]. The histological subtypes of lung cancer are divided into two main categories: small cell lung cancers and NSCLCs. NSCLC includes numerous types of lung cancer, including squamous cell carcinomas and adenocarcinomas [99].

Smoking is the main cause of lung cancer, yet lung cancer is also affected by other factors [98]. Although early studies have shown that NGF is expressed in lung cancer [100], few data on its clinicopathological significance are available. NGF and TrkA are necessary for the development of the nervous system. They stimulate the growth of sympathetic and sensory neurons [9]. Previous studies have shown that the expression of TrkA and NGF is high in NSCLC, but the disposition of TrkA and NGF in various subtypes of lung cancer is still unclear [101]. Additionally, it was found that the levels of TrkA and NGF in squamous cell carcinoma were upregulated, indicating that lung cancer drugs that target the NGF–TrkA pathway could be used for this disease (Figure 3) [102,103]. Meanwhile, in tumor tissues, the overexpression of NGF and hypoxia-inducible factor-1α (HIF-1α) boosts angiogenesis [104,105]. A recent study has also found that compared with para-cancerous lung tissues, NGF and HIF-1α are highly expressed in NSCLC tissues [20]. The expression of NGF and HIF-1α [106] is intently correlated with the microvascular density in NSCLC. This information suggests that NGF and HIF-1α possibly play considerable roles in the angiogenesis of NSCLC, which are consistent with previous findings.

These results suggest that NGF may play important roles in lung cancer and indicate that NGF provides a novel strategy to treat lung cancer.

## 4. Perspective

The lung is a very important respiratory organ. When the lungs are damaged, many respiratory diseases can occur. Although some current treatments improve some symptoms, many diseases cannot be cured. Studies have found that NGF participates in the occurrence and development of lung diseases (Figure 2) and may serve as a therapeutic target. COVID-19 still rapidly sweeps the world. Anti-NGF treatment may also improve COVID-2019. Furthermore, a few signaling pathways have been mentioned (Figure 3). In asthma, NANM significantly attenuates airway remodeling, which is related to the TGF-β1–SMAD3 signaling pathway. Anti-NGF improves AHR and relieves asthma attacks in mice by downregulating the RhoA pathway. Meanwhile, NGF could aggravate asthma by increasing the level of MMP-9. There is a relationship between NGF and TIMP-1 because of the high expression of tissue inhibitors of NGF and TIMP-1 and the correlation between these parameters in asthma patients. Anti-NGF activated the RhoA–ROCK pathway to improve PH. Simultaneously, the expression and activation of TrkA and p75NTR are increased in PH. TrkA and p75NTR are also highly expressed in COPD. TrkA is easily detected in manifold human lung cancers and fibroblasts. Therefore, these factors seem to be helpful in exploration of the mechanisms by which NGF affects the occurrence and development of lung disease. Future studies are necessary to elucidate the mechanisms underlying the effects of NGF.

## Figures and Tables

**Figure 1 ijms-22-09112-f001:**
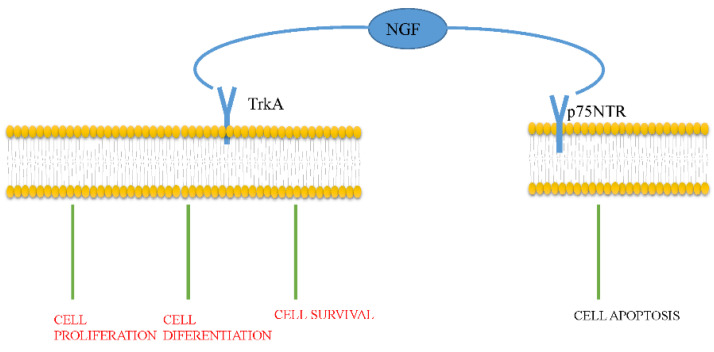
NGF exerts its function by activating TrkA and p75NTR. Activation of TrkA induces differentiation, proliferation, and survival of the target cell, while p75NTR activation induces apoptosis.

**Figure 2 ijms-22-09112-f002:**
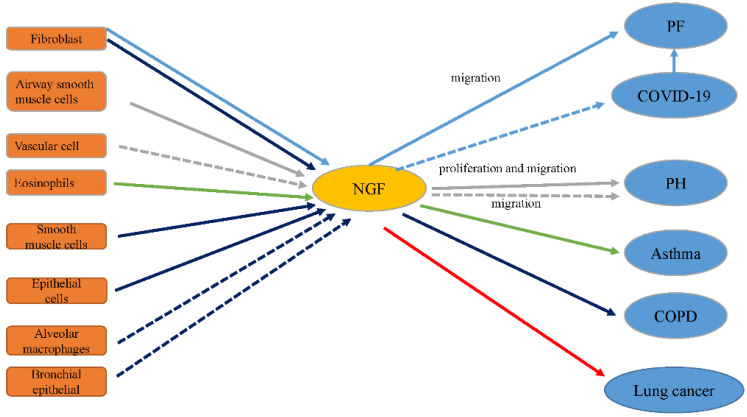
The influence of NGF on lung disease. NGF is expressed in many cells in the lung. Once released, NGF could stimulate diverse types of cells involved in many processes in lung diseases. The relationships between NGF and respective lung diseases are described in the following sections. The blue arrows represent the relationship between NGF and PF; the blue dotted arrow represents a potential effect; the gray arrows represent the relationship between NGF and PH; the green arrows indicate the relationship between NGF and asthma; the purple arrows represent the relationship between NGF and COPD; and the red arrows represent the relationship between NGF and lung cancer.

**Figure 3 ijms-22-09112-f003:**
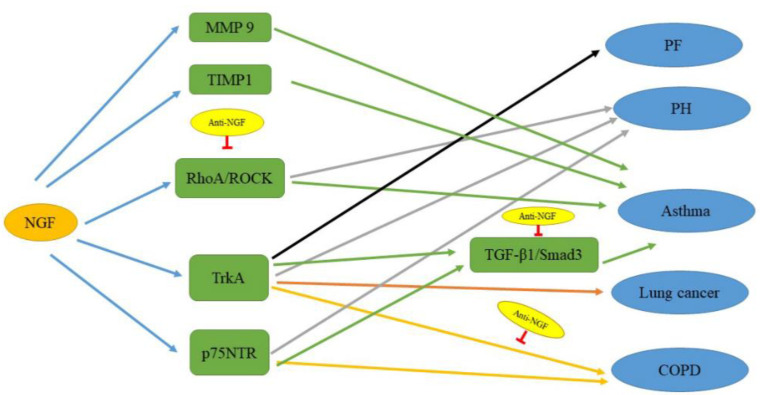
The signaling pathways related to NGF in lung diseases. NGF upregulates these factors to cause the development of lung diseases. Anti-NGF, which regulates the RhoA–ROCK, TGF-β1–SMAD3, and NGF–TrkA pathways, can improve those lung diseases. The black line indicates the pathway affecting PF; the gray lines indicate pathways affecting PH; the green lines indicate pathways affecting asthma; the orange lines indicate pathways affecting lung cancer; and the yellow lines indicate pathways affecting COPD.

## Data Availability

Not applicable.

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
