# Peer review of "Nerve Growth Factor: A Potential Therapeutic Target for Lung Diseases"

_ijms, 2021, doi:10.3390/ijms22179112_

Round 1
Reviewer 1 Report
Dear authors,
I reviewed your manuscript titled "Nerve growth factor: A potential therapeutic target for lung diseases". In my opinion the topic might be interesting to the readers, but the draft needs to be modified and improved to be acceptable.
Firt of all, it requires an extensive English revision; the style needs to be improve because it is hard to read and follow the ideas; in addition the manuscript is full of typos, doble spaces, or missing spaces, italics used sometimes to indicate figures and sometimes not, etc.
Chapter 2.2: NGF in the lung needs to be deleted or modified, because the authors try to introduce several diseases in a short paragraph, what it is unnecessary, because they will afterwards review the role of NGF in each disease. So, this paragraph can be eliminated or a better option to be used to explain what is the role of NGF in the lung in healthy conditions. Which cells are expressing NGF in a healthy lung? is detectable in the broncoalveolar lavage or in the parenchyma? Is modulated by age or sex?
Additionally, if the authors want to review different pathologies and how NGF is regulated in that disease, they need to properly describe the disease. Please, use 3-4 sentences to introduce how the disease is induced, which factors can trigger it, how it progress and then explain what is the role of NGF in each of this steps. If NGF it is up or down regulated, where, which cells, which pathways is NGF targeting in each disease and if any treatment blocking or promoting NGF was used to treat that disease.
Figure 2 is not adding much value to this review, because many of the factors that the authors include are common to different diseases. For example, why NGF is regulating inflammation in COPD and not in PF?
Figure 3 is nice and it might be better to add more information to this figure, pointing how NGF regulates each pathway and if the up or down-regulation of the pathway is inducing or avoiding disease progression.
Author Response
Dear Editor and Reviewers,
We thank reviewers for their critical and valuable comments on our manuscript. We have revised the manuscript as suggested.
- Comment: First of all, it requires an extensive English revision; the style needs to be improve because it is hard to read and follow the ideas; in addition the manuscript is full of typos, doble spaces, or missing spaces, italics used sometimes to indicate figures and sometimes not, etc. 

Response: We are very sorry for the reading trouble in the review, and apologise for the poor grammar and language issues. We have sent the manuscript to Letpub language editing company (United States) to improve the quality of the language. Enclosed is the certification of language editing. Furthermore, we have double-checked the spelling and formatting errors all through the whole manuscript. And all spelling and formatting errors have been revised according to the comments, and the text was highlighted in the revised manuscript.
- Comment: Chapter 2.2: NGF in the lung needs to be deleted or modified, because the authors try to introduce several diseases in a short paragraph, what it is unnecessary, because they will afterwards review the role of NGF in each discease. So, this paragraph can be eliminated or a better option to be used to explain what is the role of NGF in the lung in healthy conditions. Which cells are expressing NGF in a healthy lung? is detectable in the broncoalveolar lavage or in the parenchyma? Is modulated by age or sex?
Response: Thanks for the constructive suggestions and we have re-designed the structure of this part. According to your suggestion, we have described the role of NGF in the lung in healthy conditions, and the expressing level of NGF in different lung cells. Meanwhile, we show whether NGF may be detectable in the bronchoalveolar lavage or the parenchyma in Lines 141-148. We modify NGF in the lung in Lines 149-154. And whether NGF could be modulated by age or gender in Lines 226. Finally, we provide the question that the role of NGF in lung diseases. The related texts were highlighted in the revised manuscript.
- Comment: Additionally, if the authors want to review different pathologies and how NGF is regulated in that disease, they need to properly describe the disease. Please, use 3-4 sentences to introduce how the disease is induced, which factors can trigger it, how it progress and then explain what is the role of NGF in each of this steps. If NGF it is up or down regulated, where, which cells, which pathways is NGF targeting in each disease and if any treatment blocking or promoting NGF was used to treat that disease.
Response: We gratefully appreciate for your valuable suggestion. We have added some sentences to introduce how the disease is induced, which factors can trigger it, how it progresses in Lines 247-250, 274-275, 380-382, 404-406,730-732, 1056-1058,1310-1311 in the revised manuscript. Meanwhile, the role of NGF in each disease has been shown in revised Figure 3. Besides, the cells relate to NGF have also displayed in revised Figure 2 and the related pathways are targeted as well as treatment blocking or promoting NGF used to treat that disease have been described in revised Figure 3.
- Comment: Figure 2 is not adding much value to this review, because many of the factors that the authors include are common to different diseases. For example, why NGF is regulating inflammation in COPD and not in PF?
Response 4: In the previous Figure 2, it seems that NGF is only described to regulate inflammation in COPD and not in PF. We are very sorry to make the reviewer confused because we missed some information and thank you for your careful check. We have revised Figure 2 to make the information look more organized. Meanwhile, we added some missing information in the revised Figure2.
- Comment: Figure 3 is nice and it might be better to add more information to this figure, pointing how NGF regulates each pathway and if the up or down-regulation of the pathway is inducing or avoiding disease progression.
Response: Thank you for useful suggestion. We have added related missing information to this figure, which pointing how NGF regulates each pathway and the role and expressing level of NFG. More details were described to the topic, and related text was highlighted in the manuscript.
Again, we thank you for the useful comments and suggestions. We deeply appreciate your consideration of our manuscript. If you have any queries, please don’t hesitate to contact me at the address below.
Thank you and best regards.
Sincerely yours,
Liling Tang
PhD, Professor
Reviewer 2 Report
Dear Authors,
This is an interesting review by Piaoyang Liu et al, on “Nerve growth factor: A potential therapeutic target for lung diseases”.
However, it can be improved.
- Abstract: The first 3 lines, seem a little bit immature. Please rephrase them in a more scientific language. Line 20: Change merging to emerging.
- Introduction: The same applies for the introduction (Lines 24-28). Please rephrase. In line 31 you must repeat what NGF is and end-up with the abbreviation.
- In 2. NGF: You must add a paragraph of the affected signaling pathways, since you need it for the future perspectives.
- Lines 115-119: This is all very speculative. It must be rephrased and presented more delicately.
- Lines 233-234: Rephrase.
- Lines 274-276: There is a very recent reference missing.
- Although the literature is still scarce, MicroRNAs must be added in this review, to enhance its value and visibility. Either as a separate section, or inside the relevant text concerning the affected lung conditions. This will also implement your proposal on NGF as a therapeutic target, since they are affecting major signaling pathways.
Author Response
Dear Editor and Reviewers,
We thank reviewers for their critical and valuable comments on our manuscript. We have revised the manuscript as suggested.
- Comment: Abstract: The first 3 lines, seem a little bit immature. Please rephrase them in a more scientific language. Line 20: Change merging to emerging.
Response: We sorry for this linguistic issue and spelling mistakes. The term has been replaced by specific and professional words according to the suggestion. Lines 20. have been rephrased in the revised manuscript. Meanwhile, we have double checked the spelling errors all through the manuscript and incorrect writing already have been revised. Related text has been highlighted in the revised version.
- Comment: Introduction: The same applies for the introduction (Lines 24-28). Please rephrase. In line 31 you must repeat what NGF is and end-up with the abbreviation.
Response: Thank you for the useful suggestions. Lines 24-28 have been rephrased in the revised manuscript. Meanwhile, the full descriptions of the abbreviations NGF have been supplemented in Line 51. Related text has been highlighted in the revised manuscript.
- Comment: In 2. NGF: You must add a paragraph of the affected signaling pathways, since you need it for the future perspectives.
Response: Thank you for your valuable suggestion. We have re-designed the structure of this part and added a paragraph to introduce the related pathways are targeted in Line 226-240 in the revised manuscript. We also re-designed Figure 3 according to your suggestion, some missing information have been described in new Figure 3. The related texts were highlighted in the revised version.
- Comment: Lines 115-119: This is all very speculative. It must be rephrased and presented more delicately.
Response: We are sorry for the inappropriate description. These sentences have been rephrased in Lines 373-377 in the revised manuscript.
- Comment: Lines 233-234: Rephrase.
Response: Thank you for your useful suggestion, and the sentences have been rephrased in Lines 1055-1056 in the revised manuscript.
- Comment: Lines 274-276: There is a very recent reference missing.
Response: Thank you for your careful check. The missing literature has been cited in Lines 1319-1321 of the revised manuscript and also added in the Reference list in No. [107].
- Comment: Although the literature is still scarce, MicroRNAs must be added in this review, to enhance its value and visibility. Either as a separate section, or inside the relevant text concerning the affected lung conditions. This will also implement your proposal on NGF as a therapeutic target, since they are affecting major signaling pathways.
Response: We are sorry for the missing information. We have added related MicroRNAs details in revised manuscript. And the references also added in the list at No.226-240.
Again, we thank you for the useful comments and suggestions. We deeply appreciate your consideration of our manuscript. If you have any queries, please don’t hesitate to contact me at the address below.
Thank you and best regards.
Sincerely yours,
Liling Tang
PhD, Professor
Round 2
Reviewer 1 Report
The authors improved the review following all my comments and suggestions and the English language and style is also correct and readable now. I do not have any further comment.
Reviewer 2 Report
Dear Authors,
I very much enjoyed reading the revised version of your manuscript.
I feel that in its current form, it will be a valuable addition on the existing literature.
Therefore, I strongly recommend its publication.